# Rhodium(III)-Catalyzed [4+2] Annulation via C-H Activation: Synthesis of Multi-Substituted Naphthalenone Sulfoxonium Ylides

**DOI:** 10.3390/molecules24101884

**Published:** 2019-05-16

**Authors:** Xiaohan Song, Xu Han, Rui Zhang, Hong Liu, Jiang Wang

**Affiliations:** 1State Key Laboratory of Drug Research and CAS Key Laboratory of Receptor Research, Shanghai Institute of Materia Medica, Chinese Academy of Sciences, Shanghai 201203, China; xiaohanpharm@163.com (X.S.); dtchx_cbdml@hotmail.com (X.H.); ruipharmacy@163.com (R.Z.); 2University of Chinese Academy of Sciences, No.19A Yuquan Road, Beijing 100049, China

**Keywords:** rhodium(III), sulfoxonium ylides, naphthols

## Abstract

A convenient Rh(III)-catalyzed C-H activation and cascade [4+2] annulation for the synthesis of naphthalenone sulfoxonium ylides has been developed. This method features perfect regioselectivity, mild and redox-neutral reaction conditions, and broad substrate tolerance with good to excellent yields. Preliminary mechanistic experiments were conducted and a plausible reaction mechanism was proposed. The new type naphthalenone sulfoxonium ylides could be further transformed into multi-substituted naphthols, which demonstrates the practical utility of this methodology.

## 1. Introduction

Substituted naphthols have been characterized as crucial organic motifs and are embedded in various pharmaceuticals and natural products such as rifampicin [1,2,3], gossypol [4,5,6], dioncophylline A [7,8,9], propranolol [10,11,12,13], and naftopidil [14,15,16] (Figure 1). As a result, the development of efficient methods to synthesize multi-substituted naphthols is important [17,18,19,20]. Over the past few years, transition-metal-catalyzed C–H activation has been demonstrated to be a convenient strategy to establish aromatic and heteroaromatic skeletons [21,22,23,24]. Nevertheless, the synthetic approach for multi-substituted naphthols is scarcely reported [25,26,27,28,29]. For example, it can be synthesized by the Rh(III)-catalyzed cross-coupling of benzoylates with diphenylacetylene (Scheme 1a) [25,26,27]. Recently, Li and co-workers have demonstrated a strategy using phosphonium ylides and diazo compounds to access naphthol derivatives [29]. Thus, development of an efficient, straightforward route to the naphthol framework is highly desired.

Recently, sulfoxonium ylides have been identified as a precursor of carbenoid in the transition-metal-catalyzed reactions [30,31,32]. Being successfully applied to the multi-kilogram synthesis of drug intermediates via Ir(I)-catalyzed reactions in industry [33,34] sulfoxonium ylides have also been widely investigated in the Rh(III)- [35,36,37,38,39,40,41,42,43,44], Co(III)- [45], or Ru(II)-catalyzed [46,47] C-H bond functionalization. However, the application of sulfoxonium ylides is severely limited by its substrate scope because the C1 position substitution in the ylide center is only H. To overcome such limitations, Bayer and co-workers reported the synthesis of bis-substituted sulfoxonium ylides via rhodium-catalyzed coupling of iodonium ylides with sulfoxides (Scheme 1b) [48]. Burtoloso et al. described another strategy to access *α*-aryl-*β*-keto sulfoxonium ylides using aryne [49]. Furthermore, Aïssa et al. developed a palladium-catalyzed C−H cross-coupling of *α*-ester sulfoxonium ylides with aryl halide to afford the (hetero)aryl-substituted sulfoxonium ylides, which expanded the scope of the substitution in the ylide center [50]. However, the synthetic approach for cyclic sulfoxonium ylides remains unexplored.

A seminal work reported by Li and co-workers revealed that sulfoxonium ylides could serve as weak directing-groups to participate in C-H activation [27,51]. Inspired by the previous work, we report a Rh(III)-catalyzed C-H activation and [4+2] annulation to afford the naphthalenone sulfoxonium ylides and its synthetic utility is further demonstrated through simple reactions to access multi-substituted naphthols. It is worth mentioning that, during our submission, Fan’s group also reported a very similar approach to the synthesis of naphthalenone sulfoxoniums [52].

## 2. Results and Discussion

We initiated our studies with model substrates sulfoxonium ylide **1a** and diazo compound **2a** to investigate the optimal reaction conditions (Table 1). Initially, transition-metal catalysts (Ru(II), Co(III), Ir(III), and Rh(III)), which could potentially trigger the cross-coupling of **1a** with **2a**, were screened to demonstrate the feasibility of this method (entries 1–4). To our delight, the target molecule naphthalenone sulfoxonium ylide **3aa** could be obtained in a moderate yield of 65% in the presence of [Cp*RhCl_2_]_2_ and AgSbF_6_ under air condition at r.t. for 12 h. Several typical additives, including PivOH, CsOAc, Zn(OTf)_2_, Cu(OAc)_2_, and Zn(OAc)_2_, were subsequently explored (entries 5–9), and Zn(OAc)_2_ exhibited the best additive for this annulation, because a more powerful catalyst Cp*Rh(OAc) could be formed after adding Zn(OAc)_2_ [42], while CsOAc and Zn(OTf)_2_ could not afford compound **3aa** at all. Subsequent Ag salt screening revealed that replacement of AgSbF_6_ by AgNTf_2_ decreased the yield (entry 10). Encouraged by these results, we further screened the solvent and found that TFE, MeOH, and MeCN reduced the reaction conversion (entries 11–13). The optimal results could be achieved when sulfoxonium ylide (**1a**, 0.2 mmol) and diazo compounds (**2a**, 0.44 mmol) were treated with the catalytic system of [Cp*RhCl_2_]_2_ (5 mol%), AgSbF_6_ (30 mol%), and Zn(OAc)_2_ (30 mol%) in DCE at room temperature for 12 h.

With the optimal reaction conditions in hand, we started to explore the generality and scope of sulfoxonium ylides (**1a**–**1j**) by performing the annulation with diazo compound **2a** (Scheme 2). It was found that this reaction could tolerate various substrates with both electron-donating and electron-withdrawing substituents in the sulfoxonium ylides system, and afforded the corresponding naphthalenone sulfoxonium ylides in good to excellent yields (**3aa**–**3da**, 44–96%). Generally, sulfoxonium ylides with electron-donating substituents gave higher yields compared with electron-withdrawing substituents. To further investigate the effect of substituted group of the sulfoxonium yield, several moieties were independently introduced at the para-position of the phenyl ring while the ortho-position was blocked by chlorine. As a result, the naphthalenone sulfoxonium ylides were obtained in good to excellent yields (**3ea**–**3ia**, 78–96%). Introducing substituents at the meta-position resulted in excellent yields (**3ja**–**3la**, 84–94%). It is worth noting that using ortho-non-substituted benzoyl sulfoxonium ylides (**1m**–**1p**) with **2a**, the dialkylated product could be obtained in good yields (**3ma**–**3pa**, 59–76%).

Next, in order to expand the utility of this reaction, we investigated the scope and generality of the diazo compounds (Scheme 3). Diazo compounds with the electron-donating and halogen groups at the para-position of its phenyl ring (R^2^) resulted in good to excellent yields of corresponding products (**3ab**–**3ad** and **3af**, 79–95%), while electron-withdrawing group led to poor yield (**3ae**, 53%). The structure of product **3ac** was confirmed by X-ray crystallography (CCDC 1899265). It should be mentioned that the substituents of diazo compounds at the different positions of its phenyl ring (R^2^) did not alter the reaction efficiency, provided the desired products in high yields (**3ag**–**3aj**, 77%–84%). Moreover, when R^2^ was replaced by methyl or cyclopropyl the yields are 91% and 72%, respectively (**3ak** and **3al**), which indicated that increasing of the steric hindrance of R^2^ group decreased the yield of this reaction. At the same time, R^3^ groups with the large steric hindrance were well tolerated in this reaction (**3am** and **3an**, 89% and 61%).

To further assess synthetic utility of the reaction, a gram-scale reaction between **1a** and **2a** has been performed, and the product **3aa** was isolated with a 79% yield (Scheme 4a). Moreover, as a versatile structural motif, the synthetic application of the naphthalenone sulfoxonium ylides has been investigated. Naphthalenone sulfoxonium ylide **3ak** was transformed to the tetra-substituted α-naphthol **5ak**, of which the skeleton was embedded in rifampicin [1,2,3], via Ir(II)-catalyzed amination in a moderate yield of 45% (Scheme 4b) [49]. In addition, compound **3ak** was reduced to sulfoxide **6ak** in a good yield of 65%, which could be used to synthesize the FabH inhibitor [51,53], (Scheme 4c).

To obtain more insight into the mechanism of this annulation, a series of experiments were performed (Scheme 5). First, a hydrogen−deuterium exchange experiment of **1a** was carried out using CD_3_OD under the standard conditions (Scheme 5a). Compound **1a** underwent slight H/D exchange in the presence of the Rh(III) catalyst, indicating the reversibility of the C(aryl)−H bond cleavage. To further probe the C-H activation process, the kinetic isotopic effect (KIE) studies with separate kinetic experiments were performed to gain insights into the rate-determining step for this cross-coupling reaction (Scheme 5b). The KIE was determined by performing intermolecular competition experiments using an equimolar mixture of **1a** and **1a-d_7_** in the couplings with **2k** under standard conditions. The KIE value was 2.8, which was observed on the basis of the ^1^H NMR analysis (see Appendix A), indicating that the C–H activation was involved in the turnover-limiting step.

Based on these preliminary mechanistic investigations, a plausible reaction mechanism for the formation of naphthalenone sulfoxonium ylide **3aa** is proposed in Scheme 6. Initially, oxygen coordination of **1a** is followed by cyclometalation to deliver a five-membered rhodacyclic intermediate **A**. Then, the nucleophilic C(aryl)−Rh species further attacks the diazo compound **2a** to generate Rh(III) carbene species **B** with the loss of N_2_. The resulting species **B** further undergoes carbene migratory insertion to furnish another six-membered rhodacyclic intermediate **C**. Protonolysis of the Rh−C bond by HX releases the key intermediate **D** with the regeneration of the active Rh(III) catalyst. Finally, intermediate **D** undergoes a sequential aldol condensation to form the desired product **3aa**.

## 3. Materials and Methods

### 3.1. General Information

The reagents (chemicals) were purchased from commercial sources, and used without further purification. Analytical thin layer chromatography (TLC) was HSGF 254 (0.15–0.2 mm thickness). All products were characterized by their NMR and MS spectra. The ^1^H- (500 MHz) and ^13^C-NMR (125 MHz) spectra were recorded in deuterochloroform (CDCl3) on Bruker Avance III spectrometer (Billerica, MA, USA). Chemical shifts were reported in parts per million (ppm, δ) downfield from tetramethylsilane. Proton coupling patterns are described as singlet (s), doublet (d), triplet (t), quartet (q), or multiplet (m). Low-resolution mass spectra (LRMS) were measured on Agilent 1260 Infinity II (Palo Alto, CA, USA). High-resolution mass spectra (HRMS) were measured on Agilent 1290-6545 UHPLC-QTOF respectively (Palo Alto, CA, USA).

### 3.2. Experimental Part Method

#### 3.2.1. General Procedure for the Preparation of Sulfoxonium Ylides **1a**–**1p**

Sulfoxonium ylides **1a**–**1p** were prepared according to the reported procedures [28]. To a stirred solution of potassiumtert-butoxide (3.3 equiv.) in THF was added trimethylsulfoxonium iodide (3.0 equiv.) at room temperature. The resulting mixture is refluxed for 2 h. Then reaction mixture was cooled to 0 °C, followed by the addition of acyl chlorides (1.0 equiv.) in THF. The reaction was allowed to reach room temperature and stirred for 3 h. Next, the solvent was evaporated, and water and ethylacetate were added to the resulting slurry. The layers were separated and the aqueous layer was washed with ethyl acetate and the organic layers were combined. The organic solution was dried over anhydrous sodium sulphate (Na_2_SO_4_), filtered over a sintered funnel, and evaporated to dryness. The crude product was purified by flash chromatography over silica gel using DCM/MeOH (95:5) to afford the corresponding sulfoxonium ylides **1a**–**1p**.

#### 3.2.2. General Procedure for the Preparation of α-Diazocarbonyl Compounds **2a**–**2n**

The α-diazocarbonyl compounds **2a**–**2n** were prepared according to the reported procedures [29]. To a solution of β-ketoester or β-diketone (1.0 equiv.) and *N*-(4-azidosulfonylphenyl)acetamide (1.2 equiv.) in CH_3_CN at 0 °C was added DBU (1.2 equiv.). The resulting solution was stirred at 0 °C for 3 h and slowly brought to room temperature. Upon completion, as indicated by thin layer chromatography (TLC), the reaction was quenched with water, extracted with ethyl acetate, and dried over anhydrous Na_2_SO_4_. The reaction mixture was concentrated under reduced pressure, and the crude product was purified by column chromatography using *n*-hexane/EtOAc (92:8) to afford corresponding α-diazocarbonyl compounds **2a**–**2n**.

#### 3.2.3. General Procedures for the Products **3aa**–**3la**, **3ab**–**3an** (Compound **3aa** as the Example)

A tube was charged with [Cp*RhCl_2_] _2_ (6.0 mg, 5 mol%), AgSbF_6_ (14 mg, 20 mol%), Zn(OAc)_2_ (14 mg, 30 mol%), sulfoxonium ylide (1a, 0.2 mmol), α-diazocarbonyl compound (2a, 0.24 mmol), and DCE (3 mL). The reaction mixture was stirred at room temperature for 12 h under air condition. After that, the solvent was removed under reduced pressure and the residue was purified by silica gel chromatography using DCM/MeOH (98:2) to afford the product **3aa** as a light yellow solid.

#### 3.2.4. General Procedures for the Products **3ma**–**3pa** (Compound **3ma** as the Example)

A tube was charged with [Cp*RhCl_2_] _2_ (6.0 mg, 5 mol%), AgSbF_6_ (14 mg, 20 mol%), Zn(OAc)_2_ (14 mg, 30 mol%), sulfoxonium ylide (1m, 0.2 mmol), α-diazocarbonyl compound (2a, 0.44 mmol), and DCE (3 mL). The reaction mixture was stirred at 60 °C for 4 h under air condition. After that, the solvent was removed under reduced pressure and the residue was purified by silica gel chromatography using DCM/MeOH (98:2) to afford the product **3ma** as a light yellow solid.

#### 3.2.5. Gram-Scale Synthesis of Compound **3aa**


A round bottomed flask was charged with [Cp*RhCl_2_]_2_ (147 mg, 238 μmol), AgSbF_6_ (327 mg, 951 μmol), Zn(OAc)_2_ (262 mg, 1.43 mmol), sulfoxonium ylide (1a, 4.76 mmol), α-diazocarbonyl compound (2a, 1.25 g, 5.71 mmol). Dichloroethane (35 mL) was then added to the reaction mixture and stirring was turned on. The reaction mixture was stirred at r.t. for 12 h under air condition. After that, the solvent was removed under reduced pressure and the residue was purified by silica gel chromatography using DCM/MeOH (99:1) to afford the product **3aa** (1.45 g, 79%, light yellow solid).

#### 3.2.6. Synthesis of Compound **5ak**

To a 15 mL microwave glass tube containing a magnetic stirrer and fitted with a Teflon cap, sulfoxonium ylide 3ak (64 mg, 1.0 equiv.), *p*-methoxyaniline 4 (24 mg, 2.0 equiv.), [Ir(COD)Cl]_2_ (3 mg, 2.5 mol%), and toluene (1 mL) were added. The mixture was stirred for 1 h at 150 °C under microwave irradiation. Then, the organic solvent was removed in a rotary evaporator and the crude product purified by flash chromatography (petroleum ether: ethyl acetate = 10:1).

#### 3.2.7. Synthesis of Compound **6ak**

A mixture of **3ak** (64 mg, 1 equiv.) and NaH (60%, dispersion in paraffin liquid) (28 mg, 0.7 mmol, 3.5 equiv.) was added to a Schlenk tube equipped with a stir bar. Dry THF (1.0 mL) was added and the mixture was stirred at 80 °C for 24 h under Ar atmosphere. Then, the organic solvent was removed in a rotary evaporator and the crude product was purified by flash chromatography (petroleum ether: ethyl acetate = 10:1). 

#### 3.2.8. Mechanistic Studies

A tube was charged with [Cp*RhCl_2_] _2_ (6.0 mg, 5 mol%), AgSbF_6_ (14 mg, 20 mol%), Zn(OAc)_2_ (14 mg, 30 mol%), sulfoxonium ylide (1a, 0.2 mmol), CD_3_OD (72 mg, 10 equiv.), and DCE (3 mL). The reaction mixture was stirred at r.t. for 12 h under air condition. After that, the solvent was removed under reduced pressure and the residue was purified by silica gel chromatography using DCM/MeOH (96:4) to afford the product, which was characterized by ^1^H NMR spectroscopy. ^1^H NMR analysis of **1a** revealed 47% deuteration at the 6-position of phenyl ring and 8% deuteration at the α-position of the carbonyl.

Two tubes were charged with [Cp*RhCl_2_]_2_ (6.0 mg, 5 mol%), AgSbF_6_ (14 mg, 20 mol%), Zn(OAc)_2_ (14 mg, 30 mol%), sulfoxonium ylide (**1a** or **1a–*d_7_***, 0.2 mmol), α-diazocarbonyl compounds (**2k**, 0.24 mmol) and DCE (3 mL). The reaction mixture was stirred at r.t. for 2 h under air condition. After that, the solvent was removed under reduced pressure and the residue was purified by silica gel chromatography using DCM/MeOH (99:1) to afford the product. The KIE value was determined to be k_H_/k_D_ = 2.8 on the basis of ^1^H NMR analysis.

### 3.3. Product Characterization

Ethyl 3-(dimethyl(oxo)-λ^6^-sulfanylidene)-5-methyl-4-oxo-2-phenyl-3,4-dihydronaphthalene-1-carboxylate (**3aa**): light yellow solid; m.p.:182–184 °C; ^1^H NMR (400 MHz, Chloroform-d) δ 7.51 (d, *J* = 8.3 Hz, 1H), 7.43 (dd, *J* = 8.3, 7.1 Hz, 1H), 7.38–7.31 (m, 5H), 7.17 (d, *J* = 7.1, 1H), 3.92 (q, *J* = 7.1 Hz, 2H), 3.77 (s, 6H), 2.99 (s, 3H), 0.90 (t, *J* = 7.1 Hz, 3H). ^13^C NMR (125 MHz, Chloroform-d) δ 176.1, 169.3, 139.5, 137.3, 136.9, 135.9, 130.2, 129.2, 128.8, 128.6, 127.4, 127.2, 123.1, 118.4, 98.3, 60.8, 44.2, 24.4, 13.7. LRMS (ESI): 381.4 [M − H]^+^. HRMS (ESI) calculated for C_21_H_20_O_4_S [M − H]^+^: 381.1166; found: 381.1177.

Ethyl 5-chloro-3-(dimethyl(oxo)-λ^6^-sulfanylidene)-4-oxo-2-phenyl-3,4-dihydronaphthalene-1-carboxylate (**3ba**): light yellow solid; m.p.: 225–226 °C; ^1^H NMR (500 MHz, Chloroform-d) δ 7.57 (dd, *J* = 6.4, 3.1 Hz, 1H), 7.41–7.37 (m, 3H), 7.35–7.31 (m, 3H), 7.31–7.26 (m, 2H), 3.90 (q, *J* = 7.1 Hz, 2H), 3.79 (s, 6H), 0.88 (t, *J* = 7.1 Hz, 3H). ^13^C NMR (125 MHz, Chloroform-d) δ 172.7, 168.3, 138.1, 136.7, 135.7, 132.6, 129.9, 128.6, 128.2, 127.2, 126.8, 125.6, 123.7, 117.4, 99.1, 60.5, 43.9, 13.1. LRMS (ESI): 403.3 [M − H]^+^. HRMS (ESI) calculated for C_21_H_19_ClO_4_S [M − H]^+^: 403.0765; found: 403.0774.

Ethyl 5-bromo-3-(dimethyl(oxo)-λ^6^-sulfanylidene)-4-oxo-2-phenyl-3,4-dihydronaphthalene-1-carboxylate (**3ca**): light yellow solid; m.p.: 203-204 °C; ^1^H NMR (400 MHz, Chloroform-d) δ 7.66 (d, *J* = 7.6 Hz, 1H), 7.62 (d, *J* = 8.3 Hz, 1H), 7.42–7.23 (m, 6H), 3.89 (q, *J* = 7.1 Hz, 2H), 3.76 (s, 6H), 0.87 (t, *J* = 7.1 Hz, 3H). ^13^C NMR (125 MHz, Chloroform-d) δ 172.9, 168.7, 138.5, 137.1, 136.3, 132.5, 130.6, 129.1, 127.6, 127.2, 126.6, 124.8, 120.2, 117.7, 99.2, 61.0, 44.2, 13.6. LRMS (ESI): 447.2 [M − H]^+^. HRMS (ESI) calculated for C_21_H_19_BrO_4_S [M − H]^+^: 447.0260; found: 447.0254.

Ethyl 3-(dimethyl(oxo)-λ^6^-sulfanylidene)-4-oxo-2-phenyl-5-(trifluoromethyl)-3,4-dihydronaphthalene-1-carboxylate (**3da**): light yellow solid; m.p.: 228-230 °C; ^1^H NMR (400 MHz, Chloroform-d) δ 7.91 (d, *J* = 8.4 Hz, 1H), 7.86 (d, *J* = 7.6 Hz, 1H), 7.61 (t, *J* = 7.9 Hz, 1H), 7.41–7.30 (m, 5H), 3.98–3.85 (q, *J* = 7.2 Hz, 2H), 3.79 (s, 6H), 0.89 (t, *J* = 7.2 Hz, 3H). ^13^C NMR (100 MHz, Chloroform-d) δ 172.0, 168.8, 139.2, 136.4, 136.2, 129.6, 129.3, 129.0, 127.7 (q, *J_C_*_–*F*_ = 31.0 Hz), 127.7, 127.3, 125.2 (q, *J_C_*_–*F*_ = 8.2 Hz), 124.5 (*J_C_*_–*F*_ = 271.0 Hz), 117.4, 100.2, 61.0, 43.8, 13.6. ^19^F NMR (470 MHz, Chloroform-d) δ -56.9. LRMS (ESI): 459.2 [M − H]^+^. HRMS (ESI) calculated for C_22_H_19_F_3_O_4_S [M + Na]^+^: 459.0848; found: 459.0857.

Ethyl 5-chloro-3-(dimethyl(oxo)-λ^6^-sulfanylidene)-7-methyl-4-oxo-2-phenyl-3,4-dihydronaphthalene-1-carboxylate (**3ea**): light yellow solid; m.p.: 212–214 °C; ^1^H NMR (400 MHz, Chloroform-d) δ 7.42–7.27 (m, 6H), δ 7.24 (d, *J* = 1.6 Hz, 1H). 3.89 (q, *J* = 7.1 Hz, 2H), 3.75 (s, 6H), 2.40 (s, 3H), 0.86 (t, *J* = 7.1 Hz, 3H). ^13^C NMR (125 MHz, Chloroform-d) δ 173.0, 168.9, 141.0, 138.6, 137.1, 136.4, 132.8, 130.2, 129.1, 127.6, 127.2, 124.0, 123.7, 117.6, 99.0, 61.0, 44.4, 21.5, 13.6. LRMS (ESI): 417.4 [M − H]^+^. HRMS (ESI) calculated for C_22_H_21_ClO_4_S [M − H]^+^: 417.0922; found: 417.0927.

Ethyl 5-chloro-3-(dimethyl(oxo)-λ6-sulfanylidene)-7-fluoro-4-oxo-2-phenyl-3,4-dihydronaphthalene-1-carboxylate (**3fa**): light yellow solid; m.p.: 208–210 °C; ^1^H NMR (500 MHz, Chloroform-d) δ 7.37–7.32 (m, 3H), 7.30–7.24 (m, 3H), 7.17 (dd, *J* = 8.3, 2.5 Hz, 1H), 3.87 (q, *J* = 7.1 Hz, 2H), 3.77 (s, 6H), 0.86 (t, *J* = 7.2 Hz, 3H). ^13^C NMR (125 MHz, Chloroform-d) δ 172.2, 167.9, 161.8 (d, *J_C_*_–*F*_ = 251.6 Hz), 139.8, 137.9 (d, *J_C_*_–*F*_ = 10.4 Hz), 135.6, 134.7 (d, *J_C_*_–*F*_ = 11.8 Hz), 128.4, 127.3, 126.8, 122.6, 117.0 (d, *J_C_*_–*F*_ = 26.3 Hz), 116.8 (d, *J_C_*_–*F*_ = 3.7 Hz), 108.5 (d, *J_C_*_–*F*_ = 22.2 Hz), 99.3, 60.7, 43.9, 13.1. ^19^F NMR (470 MHz, Chloroform-d) δ −108.2. LRMS (ESI): 421.2 [M − H]^+^. HRMS (ESI) calculated for C_21_H_18_FClO_4_S [M − H]^+^: 421.0676; found: 421.0671.

Ethyl 5,7-dichloro-3-(dimethyl(oxo)-λ^6^-sulfanylidene)-4-oxo-2-phenyl-3,4-dihydronaphthalene-1-carboxylate (**3ga**): light yellow solid; m.p.: 215–217 °C; ^1^H NMR (400 MHz, Chloroform-d) δ 7.55 (d, *J* = 1.9 Hz, 1H), 7.37–7.31 (m, 4H), 7.29–7.25 (m, 3H), 3.87 (q, *J* = 7.1 Hz, 2H), 3.75 (s, 6H), 0.85 (t, *J* = 7.1 Hz, 3H). ^13^C NMR (125 MHz, Chloroform-d) δ 172.6, 168.3, 140.2, 137.7, 136.0, 135.9, 134.3, 129.0, 128.4, 127.8, 127.3, 124.4, 123.3, 116.8, 100.4, 61.2, 44.2, 13.6. LRMS (ESI): 437.2 [M − H]^+^. HRMS (ESI) calculated for C_21_H_18_Cl_2_O_4_S [M − H]^+^: 437.0382; found: 437.0376.

Ethyl 5-chloro-3-(dimethyl(oxo)-λ^6^-sulfanylidene)-7-methoxy-4-oxo-2-phenyl-3,4-dihydronaphthalene-1-carboxylate (**3ha**): light yellow solid; m.p.: 217–218 °C; ^1^H NMR (400 MHz, Chloroform-d) δ 7.35–7.25 (m, 5H), 7.03 (d, *J* = 2.5 Hz, 1H), 6.96 (d, *J* = 2.5 Hz, 1H), 3.86 (q, *J* = 7.1 Hz, 2H), 3.83(s, 3H), 3.72 (s, 6H), 0.85 (t, *J* = 7.1 Hz, 3H). ^13^C NMR (150 MHz, Chloroform-d) δ 172.7, 168.8, 160.1, 139.5, 138.3, 136.4, 134.3, 128.9, 127.5, 127.1, 120.5, 118.1, 117.1, 105.1, 98.4, 60.8, 55.4, 44.3, 13.5. LRMS (ESI): 433.3 [M − H]+. HRMS (ESI) calculated for C_22_H_21_ClO_5_S [M − H]+: 433.0871; found: 433.0874.

Ethyl 5-chloro-3-(dimethyl(oxo)-λ^6^-sulfanylidene)-4-oxo-2-phenyl-7-(trifluoro-methyl)-3,4-dihydronaphthalene-1-carboxylate (**3ia**): light yellow solid; m.p.: 212–214 °C; ^1^H NMR (400 MHz, Chloroform-d) δ 7.85 (d, *J* = 1.8 Hz, 1H), 7.57 (d, *J* = 1.8 Hz, 1H), 7.37 (dd, *J* = 4.9, 2.3 Hz, 3H), 7.32–7.25 (m, 2H), 3.91 (q, *J* = 7.0 Hz, 2H), 3.79 (s, 6H), 0.88 (d, *J* = 7.0 Hz, 3H). ^13^C NMR (125 MHz, Chloroform-d) δ 172.4, 168.1, 140.5, 136.9, 135.7, 134.4, 131.8 (q, *J* = 33.1 Hz), 128.9, 128.2 (q, *J* = 274.7 Hz) 127.9, 127.6, 127.3, 124.1 (q, *J* = 3.3 Hz), 121.3 (q, *J* = 4.3 Hz), 117.5, 101.7, 61.3, 44.1, 13.5. ^19^F NMR (470 MHz, Chloroform-d) δ −63.1. LRMS (ESI): 471.3 [M − H]^+^. HRMS (ESI) calculated for C_22_H_18_ClF_3_O_4_S [M − H]^+^: 471.0639; found: 471.0644.

Ethyl 6-bromo-5-chloro-3-(dimethyl(oxo)-λ^6^-sulfanylidene)-4-oxo-2-phenyl-3,4-dihydronaphthalene-1-carboxylate (**3ja**): light yellow solid; m.p.: 232–234 °C; ^1^H NMR (400 MHz, Chloroform-d) δ 7.74 (d, *J* = 8.9 Hz, 1H), 7.46 (d, *J* = 8.9 Hz, 1H), 7.38–7.34 (m, 3H), 7.32–7.29 (m, 2H), 3.89 (q, *J* = 7.1 Hz, 2H), 3.79 (s, 6H), 0.87 (t, *J* = 7.1 Hz, 3H). ^13^C NMR (125 MHz, Chloroform-d) δ 172.3, 168.4, 139.4, 136.0, 134.7, 132.5, 129.0, 127.8, 127.6, 127.3, 124.8, 122.7, 117.4, 100.7, 61.1, 44.3, 13.6. LRMS (ESI): 480.8 [M − H]^+^. HRMS (ESI) calculated for C_21_H_19_BrClO_4_S [M − H]^+^: 480.9870; found: 480.9866.

Ethyl 5-chloro-3-(dimethyl(oxo)-λ^6^-sulfanylidene)-6-methyl-4-oxo-2-phenyl-3,4-dihydronaphthalene-1-carboxylate (**3ka**): light yellow solid; m.p.: 230–232 °C; ^1^H NMR (400 MHz, Chloroform-d) δ 7.50 (d, *J* = 8.4 Hz, 1H), 7.42 (d, *J* = 8.4 Hz, 1H), 7.38–7.29 (m, 5H), 3.93–3.87 (q, *J* = 7.1 Hz 2H), 3.79 (s, 6H), 2.52 (s, 3H), 0.88 (t, *J* = 7.1 Hz, 3H). ^13^C NMR (125 MHz, Chloroform-d) δ 173.3, 168.9, 137.7, 136.4, 135.4, 135.1, 133.0, 132.3, 129.1, 127.5, 127.2, 126.3, 123.3, 117.7, 99.7, 60.9, 44.5, 21.1, 13.6. LRMS (ESI): 416.9 [M − H]^+^.HRMS (ESI) calculated for C_22_H_22_ClO_4_S [M − H]^+^: 417.0922; found: 417.0922.

Ethyl 5-chloro-3-(dimethyl(oxo)-λ^6^-sulfanylidene)-8-methoxy-4-oxo-2-phenyl-3,4-dihydronaphthalene-1-carboxylate (**3la**): light yellow solid; m.p.: 225–227 °C; ^1^H NMR (400 MHz, Chloroform-d) δ 7.35–7.26 (m, 6H), 6.90 (d, *J* = 8.5 Hz, 1H), 3.82 (q, 2H), 3.81 (s, 3H), 3.73 (s, 6H), 0.96 (t, *J* = 7.1 Hz, 3H). ^13^C NMR (150 MHz, Chloroform-d) δ 172.4, 169.3, 153.7, 138.1, 135.3, 129.8, 128.5, 128.2, 127.5, 127.1, 126.9, 124.6, 114.6, 111.7, 100.5, 60.5, 56.7, 44.0, 13.9. LRMS (ESI): 432.9 [M − H]^+^. HRMS (ESI) calculated for C_22_H_22_ClO_5_S [M − H]^+^: 433.0871; found: 433.0882.

Ethyl 3-(dimethyl(oxo)-λ^6^-sulfanylidene)-5-(1-ethoxy-1,3-dioxo-3-phenylpropan-2-yl)-4-oxo-2-phenyl-3,4-dihydronaphthalene-1-carboxylate (**3ma**): light yellow solid; m.p.: 88–90 °C; ^1^H NMR (400 MHz, Chloroform-d) δ 8.08–8.00 (m, 2H), 7.95 (s, 1H), 7.64 (d, *J* = 8.3 Hz, 1H), 7.55–7.30 (m, 9H), 7.17 (d, *J* = 7.3 Hz, 1H), 4.27 (q, *J* = 7.1 Hz, 2H), 3.91 (q, *J* = 7.2 Hz, 2H), 3.66 (s, 3H), 3.65 (s, 3H), 1.27 (t, *J* = 7.1 Hz, 3H), 0.89 (t, *J* = 7.1 Hz, 3H). ^13^C NMR (125 MHz, Chloroform-d) δ 195.4, 175.0, 170.4, 169.0, 137.6, 136.7, 136.6, 136.1, 133.8, 132.9, 130.2, 129.3, 129.2, 129.0, 128.5, 127.6, 127.2, 126.9, 125.3, 118.4, 99.5, 61.2, 60.9, 58.5, 44.0, 43.9, 14.2, 13.7. LRMS (ESI): 559.3 [M − H]^+^, HRMS (ESI) calculated for C_32_H_31_O_7_S [M − H]^+^: 559.1785; found: 559.1793.

Ethyl 3-(dimethyl(oxo)-λ^6^-sulfanylidene)-5-(1-ethoxy-1,3-dioxo-3-phenylpropan-2-yl)-7-methoxy-4-oxo-2-phenyl-3,4-dihydronaphthalene-1-carboxylate (**3na**): light yellow solid; m.p.: 95–97 °C; ^1^H NMR (400 MHz, Chloroform-d) δ 8.03 (d, *J* = 7.4 Hz, 2H), 7.92 (s, 1H), 7.55–7.48 (m, 1H), 7.45–7.31 (m, 7H), 7.02 (d, *J* = 2.5 Hz, 1H), 6.80 (d, *J* = 2.4 Hz, 1H), 4.27 (q, *J* = 7.1 Hz, 2H), 3.89 (q, *J* = 7.1 Hz, 2H), 3.80 (s, 3H), 3.67 (s, 3H), 3.66 (s, 3H), 1.27 (t, *J* = 7.1 Hz, 3H), 0.88 (t, *J* = 7.1 Hz, 3H). ^13^C NMR (150 MHz, Chloroform-d) δ 195.2, 174.7, 170.2, 169.2, 160.4, 138.6, 138.0, 136.8, 136.6, 135.8, 132.9, 129.2, 129.1, 129.0, 128.5, 127.6, 127.3, 121.5, 118.0, 117.9, 105.6, 98.2, 61.3, 60.9, 58.4, 55.2, 44.4, 44.2, 14.3, 13.7. LRMS (ESI): 589.0 [M − H]^+^. HRMS (ESI) calculated for C_33_H_33_O_8_S [M − H]^+^: 589.1891; found: 589.1870.

Ethyl 7-(tert-butyl)-3-(dimethyl(oxo)-l6-sulfanylidene)-5-(1-ethoxy-1,3-dioxo-3-phenylpropan-2-yl)-4-oxo-2-phenyl-3,4-dihydronaphthalene-1-carboxylate (**3oa**): light yellow solid; m.p.: 110–112 °C; ^1^H NMR (400 MHz, Chloroform-d) δ 8.05–7.98 (m, 2H), 7.93 (s, 1H), 7.56 (d, *J* = 1.8 Hz, 1H), 7.50 (t, *J* = 7.5 Hz, 1H), 7.42-7.30 (m, 7H), 7.20 (d, *J* = 1.8 Hz, 1H), 4.28 (qd, *J* = 7.1, 3.3 Hz, 1H), 3.95 (q, *J* = 7.1 Hz, 2H), 3.70 (s, 3H), 3.69 (s, 2H), 1.29 (t, *J* = 7.1 Hz, 3H), 1.24 (s, 9H), 0.96 (t, *J* = 7.1 Hz, 3H). ^13^C NMR (125 MHz, Chloroform-d) δ 195.8, 174.9, 170.4, 169.1, 152.9, 137.4, 136.9, 136.7, 135.9, 133.3, 132.6, 129.3, 129.2, 129.0, 128.4, 127.6, 127.3, 126.8, 120.9, 118.7, 98.7, 61.1, 60.8, 58.7, 44.3, 44.1, 35.0, 30.8, 14.3, 13.8. LRMS (ESI): 615.0 [M − H]^+^. HRMS (ESI) calculated for C_36_H_39_O_7_S [M − H]^+^: 615.2411; found: 615.2396.

Ethyl 7-bromo-3-(dimethyl(oxo)-l6-sulfanylidene)-5-(1-ethoxy-1,3-dioxo-3-phenylpropan-2-yl)-4-oxo-2-phenyl-3,4-dihydronaphthalene-1-carboxylate (**3pa**): light yellow solid; m.p.: 113–115 °C; ^1^H NMR (400 MHz, Chloroform-d) δ 8.03 (d, *J* = 7.7 Hz, 2H), 7.88–7.77 (m, 2H), 7.55 (t, *J* = 7.4 Hz, 1H), 7.44 (t, *J* = 7.6 Hz, 3H), 7.41–7.30 (m, 5H), 4.28 (q, *J* = 7.1 Hz, 2H), 3.91 (q, *J* = 7.2 Hz, 2H), 3.70 (s, 3H), 3.66 (s, 3H), 1.27 (t, *J* = 7.1 Hz, 3H), 0.88 (t, *J* = 7.2 Hz, 3H). ^13^C NMR (125 MHz, Chloroform-d) δ 194.5, 174.6, 169.8, 168.5, 139.0, 137.3, 136.7, 136.2, 135.8, 133.0, 130.3, 129.2, 129.1, 128.9, 128.6, 127.8, 127.7, 127.3, 125.5, 125.1. 117.5. 100.0, 61.5, 61.1, 58.1, 44.1, 14.2, 13.6. LRMS (ESI): 636.8 [M − H]^+^. HRMS (ESI) calculated for C_32_H_30_BrO_7_S [M − H]^+^: 637.0890; found: 637.0903.

Ethyl 3-(dimethyl(oxo)-λ^6^-sulfanylidene)-2-(4-fluorophenyl)-5-methyl-4-oxo-3,4-dihydronaphthalene-1-carboxylate (**3ab**): light yellow solid; m.p.: 217–218 °C; ^1^H NMR (500 MHz, Chloroform-d) δ 7.47 (dd, *J* = 8.5, 1H), 7.42 (dd, *J* = 8.3, 7.1 Hz, 1H), 7.31-7.26 (m, 2H), 7.16 (d, *J* = 7.0, 1H), 7.07–7.01 (m, 1H), 3.95 (q, *J* = 7.1 Hz, 2H), 3.75 (s, 6H), 2.97 (s, 3H), 0.96 (t, *J* = 7.2 Hz, 3H). ^13^C NMR (125 MHz, Chloroform-d) δ 175.6, 168.7, 161.8 (d, *J _C–F_* = 246.3 Hz), 139.1, 135.6, 135.3, 132.1 (d, *J _C–F_* = 3.5 Hz), 130.4 (d, *J _C–F_* = 8.0 Hz), 129.8, 128.4, 128.3, 122.6, 118.4, 113.7 (*J _C–F_* = 21.5 Hz), 97.6, 60.4, 43.9, 23.9, 13.3. ^19^F NMR (470 MHz, Chloroform-d) δ −144.6. LRMS (ESI): 401.2 [M − H]^+^. HRMS (ESI) calculated for C_22_H_21_FO_4_S [M − H]^+^: 401.1223; found: 401.1226.

Ethyl 2-(4-chlorophenyl)-3-(dimethyl(oxo)-λ^6^-sulfanylidene)-5-methyl-4-oxo-3,4-dihydronaphthalene-1-carboxylate (**3ac**): light yellow solid; m.p.: 199–201°C; ^1^H NMR (400 MHz, Chloroform-d) δ 7.45 (d, *J* = 8.2 Hz, 1H), 7.40 (t, *J* = 7.6 Hz, 1H), 7.29 (d, *J* = 8.1 Hz, 2H), 7.22 (d, *J* = 8.2 Hz, 2H), 7.14 (d, *J* = 7.0 Hz, 1H), 3.93 (q, *J* = 7.1 Hz, 2H), 3.69 (s, 6H), 2.94 (s, 3H), 0.93 (t, *J* = 7.1 Hz, 3H). ^13^C NMR (125 MHz, Chloroform-d) δ 176.0, 169.1, 139.6, 135.9, 135.8, 135.3, 133.4, 130.6, 130.3, 128.8, 127.4, 123.1, 118.7, 98.0, 60.9, 44.2, 24.3, 13.7. LRMS (ESI): 417.2 [M − H]^+^. HRMS (ESI) calculated for C_22_H_21_ClO_4_S [M − H]^+^: 417.0922; found: 417.0927.

Ethyl 2-(4-bromophenyl)-3-(dimethyl(oxo)-λ^6^-sulfanylidene)-5-methyl-4-oxo-3,4-dihydronaphthalene-1-carboxylate (**3ad**): light yellow solid, 88 mg, yield: 95%. m.p.: 217–219 °C; ^1^H NMR (400 MHz, Chloroform-d) δ 7.49–7.44 (m, 3H), 7.39 (t, *J* = 7.6 Hz, 1H), 7.19–7.08 (m, 3H), 3.92 (q, *J* = 7.1 Hz, 2H), 3.67 (s, 6H), 2.94 (s, 3H), 0.93 (t, *J* = 7.1 Hz, 3H). ^13^C NMR (125 MHz, Chloroform-d) δ 175.9, 169.0, 139.6, 135.9, 135.8, 135.8, 130.9, 130.3, 128.9, 123.1, 121.6, 118.5, 98.0, 60.9, 44.2, 24.4, 13.7. LRMS (ESI): 459.2 [M − H]^+^. HRMS (ESI) calculated for C_22_H_21_BrO_4_S [M − H]^+^: 459.0271; found: 459.0263.

Ethyl 3-(dimethyl(oxo)-λ^6^-sulfanylidene)-5-methyl-4-oxo-2-(4-(trifluoromethyl)ph-enyl)-3,4-dihydronaphthalene-1-carboxylate (**3ae**): light yellow solid; m.p.: 202–204 °C; ^1^H NMR (400 MHz, Chloroform-d) δ 7.61 (d, *J* = 8.0 Hz, 2H), 7.50 (d, *J* = 8.2 Hz, 1H), 7.45 (d, *J* = 7.8 Hz, 3H), 7.19 (d, *J* = 7.0 Hz, 1H), 3.91 (q, *J* = 7.1 Hz, 2H), 3.75 (s, 6H), 2.98 (s, 3H), 0.87 (t, *J* = 7.1 Hz, 3H). ^13^C NMR (125 MHz, Chloroform-d) δ 176.1, 168.9, 140.8, 139.7, 135.8, 135.7, 130.4, 129.7, 129.3 (q, *J_C–F_* = 92.4 Hz), 129.0, 124.2 (q, *J_C–F_* = 270.3 Hz), 124.0 (q, *J_C-F_* = 8.2 Hz), 123.2, 118.6, 97.6, 60.9, 44.2, 24.4, 13.5. ^19^F NMR (470 MHz, Chloroform-d) δ −62.4. LRMS (ESI): 451.2 [M − H]^+^. HRMS (ESI) calculated for C_23_H_21_F_3_O_4_S [M − H]^+^: 451.1185; found: 451.1184.

Ethyl 3-(dimethyl(oxo)-λ^6^-sulfanylidene)-2-(4-methoxyphenyl)-5-methyl-4-oxo-3,4-dihydronaphthalene-1-carboxylate (**3af**): light yellow solid, 78 mg, yield: 95%. m.p.: 152–154 °C; ^1^H NMR (400 MHz, Chloroform-d) δ 7.50–7.46 (m, 1H), 7.42 (dd, *J* = 8.3, 7.1 Hz, 1H), 7.25 (d, *J* = 8.6 Hz, 2H), 7.15 (dt, *J* = 7.1, 1.0 Hz, 1H), 6.90 (d, *J* = 8.6 Hz, 2H), 3.96 (q, *J* = 7.1 Hz, 2H), 3.84 (s, 3H), 3.76 (s, 6H), 2.98 (s, 3H), 0.97 (t, *J* = 7.1 Hz, 3H). ^13^C NMR (100 MHz, Chloroform-d) δ 176.1, 169.4, 158.9, 139.5, 136.9, 135.8, 130.2, 130.1, 128.9, 128.7, 128.5, 122.9, 118.8, 112.6, 98.3, 60.8, 55.2, 44.3, 24.4, 13.8. LRMS (ESI): 413.3 [M − H]^+^, HRMS (ESI) calculated for C_23_H_24_O_5_S [M − H]^+^: 413.1417; found: 413.1420.

Ethyl 3-(dimethyl(oxo)-λ^6^-sulfanylidene)-2-(3-methoxyphenyl)-5-methyl-4-oxo-3,4-dihydronaphthalene-1-carboxylate (**3ag**): light yellow solid; m.p.: 80–82 °C; ^1^H NMR (400 MHz, Chloroform-d) δ 7.49 (d, *J* = 8.2 Hz, 1H), 7.41 (dd, *J* = 8.3, 7.1 Hz, 1H), 7.30–7.21 (m, 1H), 7.14 (d, *J* = 7.1 Hz, 1H), 6.94–6.85 (m, 3H), 3.95 (q, *J* = 7.1 Hz, 2H), 3.80 (s, 3H), 3.70 (d, *J* = 2.6 Hz, 6H), 2.97 (s, 3H), 0.93 (t, *J* = 7.2 Hz, 3H). ^13^C NMR (125 MHz, Chloroform-d) δ 176.1, 169.3, 158.6, 139.5, 138.2, 137.0, 135.8, 130.2, 128.8, 128.1, 123.0, 122.0, 118.1, 115.3, 112.8, 98.3, 60.8, 55.2, 44.2, 24.4, 13.7. LRMS (ESI): 413.3 [M − H]^+^, HRMS (ESI) calculated for C_19_H_24_O_4_S [M − H]^+^: 413.1417; found: 413.1427.

Ethyl 2-(3-bromophenyl)-3-(dimethyl(oxo)-λ^6^-sulfanylidene)-5-methyl-4-oxo-3,4-dihydronaphthalene-1-carboxylate (**3ah**) light yellow solid; m.p.: 90–92 °C; ^1^H NMR (400 MHz, Chloroform-d) δ 7.49–7.43 (m, 3H), 7.43–7.37 (m, 1H), 7.25–7.16 (m, 1H), 7.16–7.11 (m, 1H), 3.94 (q, *J* = 7.1 Hz, 2H), 3.70 (s, 3H), 3.69 (s, 3H), 2.94 (s, 3H), 0.94 (t, *J* = 7.1 Hz, 3H). ^13^C NMR (125 MHz, Chloroform-d) δ 175.5, 168.5, 139.1, 138.4, 135.3, 135.1, 131.7, 129.9, 129.8, 128.5, 128.2, 127.6, 122.7, 120.7, 118.1, 97.4, 60.5, 43.8, 43.7, 23.9, 13.3. HRMS (ESI) calculated for C_22_H_21_BrO_4_S [M − H]^+^: 461.0417; found: 461.0427.

Ethyl 2-(2-chlorophenyl)-3-(dimethyl(oxo)-λ^6^-sulfanylidene)-5-methyl-4-oxo-3,4-dihydronaphthalene-1-carboxylate (**3ai**): light yellow solid; m.p.: 86–88 °C; ^1^H NMR (500 MHz, Chloroform-d) δ 7.54 (d, *J* = 8.1 Hz, 1H), 7.44–7.34 (m, 2H), 7.32–7.21 (m, 3H), 7.15 (d, *J* = 7.2 Hz, 1H), 3.96–3.82 (m, 2H), 3.81 (s, 3H), 3.71 (s, 3H), 2.97 (s, 3H), 0.90 (t, *J* = 7.1 Hz, 3H). ^13^C NMR (125 MHz, Chloroform-d) δ 175.2, 168.4, 139.1, 135.7, 135.4, 134.0, 133.9, 130.2, 129.6, 128.7, 128.5, 127.8, 125.5, 123.0, 117.5, 96.7, 60.3, 43.5, 41.6, 24.0, 13.2. HRMS (ESI) calculated for C_22_H_21_BrO_4_S [M − H]^+^: 417.0922; found: 417.0931.

Ethyl 3-(dimethyl(oxo)-λ^6^-sulfanylidene)-2-(2-methoxyphenyl)-5-methyl-4-oxo-3,4-dihydronaphthalene-1-carboxylate (**3aj**): light yellow solid; m.p.: 210–212 °C; ^1^H NMR (500 MHz, Chloroform-d) δ 7.59–7.54 (m, 1H), 7.38 (dd, *J* = 8.3, 7.2 Hz, 1H), 7.33 (td, *J* = 7.9, 1.8 Hz, 1H), 7.21 (dd, *J* = 7.4, 1.7 Hz, 1H), 7.13 (dt, *J* = 7.2, 1.1 Hz, 1H), 6.96 (td, *J* = 7.4, 1.1 Hz, 1H), 6.87 (dd, *J* = 8.3, 1.0 Hz, 1H), 3.97–3.83 (m, 2H), 3.81 (s, 3H), 3.74 (s, 3H), 3.73 (s, 3H), 2.98 (s, 3H), 0.89 (t, *J* = 7.2 Hz, 3H). ^13^C NMR (125 MHz, Chloroform-d) δ 174.8, 168.9, 157.0, 138.9, 135.8, 133.8, 129.3, 129.3, 128.7, 128.4, 128.0, 125.9, 122.7, 119.8, 117.5, 109.4, 97.9, 60.1, 55.2, 43.4, 41.3, 24.0, 13.2. LRMS (ESI): 413.3 [M − H]^+^. HRMS (ESI) calculated for C_23_H_24_O_5_S [M − H]^+^: 413.1417; found: 413.1421.

Ethyl 3-(dimethyl(oxo)-λ^6^-sulfanylidene)-2,5-dimethyl-4-oxo-3,4-dihydronaphthalene-1-carboxylate (**3ak**): light yellow solid; m.p.: 146–148 °C; ^1^H NMR (500 MHz, Chloroform-d) δ 7.37–7.32 (m, 1H), 7.28 (d, *J* = 6.2 Hz, 1H), 7.06 (d, *J* = 7.1 Hz, 1H), 4.43 (q, *J* = 7.1 Hz, 2H), 3.80 (s, 6H), 2.90 (s, 3H), 2.43 (s, 3H), 1.40 (t, *J* = 7.2 Hz, 3H). ^13^C NMR (125 MHz, Chloroform-d) δ 175.6, 170.1, 138.8, 135.8, 132.6, 129.4, 127.6, 127.5, 117.4, 98.1, 60.7, 44.2, 23.8, 16.3, 13.8. LRMS (ESI): 321.2 [M − H]^+^. HRMS (ESI) calculated for C_17_H_20_O_4_S [M − H]^+^: 321.1155; found: 321.1157.

Ethyl 2-cyclopropyl-3-(dimethyl(oxo)-λ^6^-sulfanylidene)-5-methyl-4-oxo-3,4-dihy-dronaphthalene-1-carboxylate (**3al**): light yellow solid; m.p.: 180–182 °C; ^1^H NMR (500 MHz, Chloroform-d) δ 7.47 (d, *J* = 8.3 Hz, 1H), 7.35 (dd, *J* = 8.4, 7.2 Hz, 1H), 7.07 (dt, *J* = 7.1, 1.2 Hz, 1H), 4.42 (q, *J* = 7.2 Hz, 2H), 3.81 (d, *J* = 2.4 Hz, 6H), 2.89 (s, 3H), 2.25 (tt, *J* = 8.6, 5.9 Hz, 1H), 1.41 (t, *J* = 7.2 Hz, 2H), 0.93 (dd, *J* = 8.4, 1.7 Hz, 2H), 0.64 (dd, *J* = 5.9, 1.7 Hz, 2H). ^13^C NMR (125 MHz, Chloroform-d) δ 175.6, 170.0, 139.2, 138.7, 135.7, 129.2, 128.1, 127.9, 122.1, 118.2, 99.7, 60.6, 44.0, 23.9, 13.86, 13.72, 8.3. LRMS (ESI): 331.4 [M − H]^+^. HRMS (ESI) calculated for C_19_H_24_O_4_S [M − H]^+^: 331.1010; found: 331.1011.

Isopropyl 3-(dimethyl(oxo)-λ^6^-sulfanylidene)-2,5-dimethyl-4-oxo-3,4-dihydronaphthalene-1-carboxylate (**3am**): light yellow solid; m.p.: 148–150 °C; ^1^H NMR (400 MHz, Chloroform-d) δ 7.35 (t, *J* = 7.6 Hz, 1H), 7.29 (d, *J* = 7.4 Hz, 1H), 7.05 (d, *J* = 7.0 Hz, 1H), 5.35 (hept, *J* = 5.7 Hz, 1H), 3.79 (s, 6H), 2.89 (s, 3H), 2.44 (s, 3H), 1.40 (d, *J* = 6.3 Hz, 6H). ^13^C NMR (125 MHz, Chloroform-d) δ 176.3, 170.1, 139.3, 136.3, 132.8, 129.8, 128.2, 127.9, 122.3, 118.0, 98.3, 68.7, 44.6, 24.3, 21.9, 16.6. LRMS (ESI): 335.3 [M − H]^+^. HRMS (ESI) calculated for C_18_H_22_O_4_S [M − H]^+^: 335.1312; found: 335.1308.

Tert-butyl 3-(dimethyl(oxo)-λ^6^-sulfanylidene)-2,5-dimethyl-4-oxo-3,4-dihydronaphthalene-1-carboxylate (**3an**): light yellow solid; m.p.: 152–154 °C; ^1^H NMR (400 MHz, Chloroform-d) δ 7.35 (d, *J* = 4.1 Hz, 2H), 7.05 (s, 1H), 3.80 (s, 6H), 2.88 (s, 3H), 2.45 (s, 3H), 1.63 (s, 9H). ^13^C NMR (125 MHz, Chloroform-d) δ 176.0, 169.9, 139.3, 136.4, 132.1, 129.8, 128.1, 127.9, 122.3, 119.4, 98.2, 81.7, 44.7, 28.3, 24.3, 16.5. LRMS (ESI): 349.3 [M − H]^+^. LRMS (ESI): 349.4 [M − H]^+^, HRMS (ESI) calculated for C_19_H_24_O_4_S [M − H]^+^: 349.1468; found: 349.1472.

Ethyl 4-hydroxy-3-((4-methoxyphenyl)amino)-2,5-dimethyl-1-naphthoate (**5ak**): green oil, 33 mg, yield: 35%. ^1^H NMR (400 MHz, DMSO-*d*_6_) δ 9.15 (s, 1H), 7.43 (d, *J* = 8.4 Hz, 1H), 7.34 (dd, *J* = 8.6, 6.9 Hz, 1H), 7.19 (d, *J* = 6.9 Hz, 1H), 6.96 (s, 1H), 6.74 (d, *J* = 8.9 Hz, 2H), 6.45 (d, *J* = 8.9 Hz, 2H), 4.39 (q, *J* = 7.1 Hz, 2H), 3.63 (s, 3H), 2.88 (s, 3H), 2.11 (s, 3H), 1.32 (t, *J* = 7.1 Hz, 3H). ^13^C NMR (100 MHz, DMSO-*d*_6_) δ 169.8, 154.0, 152.3, 141.6, 135.7, 133.3, 130.6, 127.8, 126.9, 123.7, 123.4, 122.6, 122.3, 115.0, 114.9, 61.3, 55.7, 25.2, 16.0, 14.5. LRMS (ESI): 364.4 [M − H]^+^, HRMS (ESI) calculated for C_22_H_23_NO_4_ [M – H]^+^: 364.1154; found: 364.1157. 

Ethyl 4-hydroxy-2,5-dimethyl-3-(methylsulfinyl)-1-naphthoate (**6ak**): white solid, 40 mg, yield: 65%. m.p.: 128–130 °C. ^1^H NMR (500 MHz, Chloroform-*d*) δ 12.33 (s, 1H), 7.53–7.48 (m, 1H), 7.43 (dd, *J* = 8.5, 7.0 Hz, 1H), 7.23 (dt, *J* = 7.0, 1.1 Hz, 1H), 4.51 (q, *J* = 7.2 Hz, 2H), 3.05 (s, 3H), 2.96 (s, 3H), 2.37 (s, 3H), 1.45 (t, *J* = 7.2 Hz, 3H). ^13^C NMR (125 MHz, Chloroform-*d*) δ 169.5, 162.9, 137.5, 133.9, 129.0, 129.0, 128.0, 124.5, 124.2, 114.5, 61.6, 39.4, 25.1, 16.0, 14.3. LRMS (ESI): 306.5 [M − H]^+^, HRMS (ESI) calculated for C_16_H_18_O_4_S [M − H]^+^: 308.0853; found: 308.0854.

## 4. Conclusions

In summary, we developed a novel method to access naphthalenone sulfoxonium ylides via Rh(III)-catalyzed C-H activation and [4+2] annulation of sulfoxonium ylides with diazo compounds. High regioselectivity, mild and redox-neutral reaction conditions, and wide substrate tolerance make this protocol efficient to prepare various naphthalenone sulfoxonium ylides. Moreover, the new type of naphthalenone sulfoxonium ylides could be further transformed into multi-substituted naphthols smoothly, which may find important applications in the synthesis of natural products and biologically-active molecules.

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
