# Peer review of "Rhodium(III)-Catalyzed [4+2] Annulation via C-H Activation: Synthesis of Multi-Substituted Naphthalenone Sulfoxonium Ylides"

_molecules, 2019, doi:10.3390/molecules24101884_

Round 1
Reviewer 1 Report
The paper describes the synthesis of multi-substituted naphthalenone sulfoxonium ylides by Rh(III)-catalyzed C-H activation and cascade [4+2] annulation of sulfoxonium ylides and diazo compounds as a carbine synthons. The reaction is novel, and the most important advantages of this paper are high regioselectivity, tolerance of functional groups, mild and redox-neutral reaction conditions and wide substrate scope. The structure of the novel compounds is well proved by NMR spectroscopy and mass-spectrometry. The paper is well and clearly written and deserves publication in Molecules after addressing the following points.
1. Low-resolution mass-spectral data were not given in the paper.
2. The quantity of the 13C NMR signals for the compound 6ak did not fit the molecular formulae.
3. The reference 52 is given in an unacceptable form.
4. The necessary data for NMR- and mass-spectrometers were not given.
5. There are some misprints throughout the paper, which should be corrected by extensive editing.
Author Response
Reviewer: 1
Comments:
The paper describes the synthesis of multi-substituted naphthalenone sulfoxonium ylides by Rh(III)-catalyzed C-H activation and cascade [4+2] annulation of sulfoxonium ylides and diazo compounds as a carbine synthons. The reaction is novel, and the most important advantages of this paper are high regioselectivity, tolerance of functional groups, mild and redox-neutral reaction conditions and wide substrate scope. The structure of the novel compounds is well proved by NMR spectroscopy and mass-spectrometry. The paper is well and clearly written and deserves publication in Molecules after addressing the following points.
Question 1: Low-resolution mass-spectral data were not given in the paper.
Response: We are very grateful to the reviewer’s comments.
We have listed the low-resolution mass-spectral data in the revised manuscript.
Question 2: The quantity of the 13C NMR signals for the compound 6ak did not fit the molecular formulae.
Response: We are very grateful to the reviewer’s comments.
We are so sorry that we made mistake of the 13C NMR spectra of 6ak. We have revised the 13C NMR spectra of 6ak in our revised manuscript.
Question 3: The reference 52 is given in an unacceptable form.
Response: We are very grateful to the reviewer’s comments.
We have revised the form of reference 52 in the revised manuscript.
Question 4: The necessary data for NMR- and mass-spectrometers were not given.
Response: We are very grateful to the referee for his/her suggestions.
We have described the NMR- and mass-spectrometers and listed the copies of high -resolution mass spectra for the products in the revised supporting information.
Question 5: There are some misprints throughout the paper, which should be corrected by extensive editing.
Response: We are very grateful to the referee for his/her suggestions.
We have revised English grammar and spelling in the revised manuscript.
Reviewer 2 Report
In the work entitled „Rhodium(III)-Catalyzed [4+2] Annulation via C-H Activation: Synthesis of Multi-substituted Naphthalenone Sulfoxonium Ylides „ by Xiaohan Song, Xu Han, Rui Zhang, Hong Liu, and Jiang Wangthe, the Authors touches an interesting method of [4+2] annulation of naphthalene derivatives on rhodium catalyst.
Generally, the work is presented clearly and contains a promising solution for obtaining substituted naphthalenes. There are also appropriate literature citations and the results are supported by relevant data.
Nevertheless I have minor remarks:
1. English revision recommended.
2. I wonder if the Authors did identify any other regioisomers in the reaction mixture among the products shown on Scheme 2. If yes, such information would be valuable in the method description.
3. Scheme 1: Abbreviations like DG, ACN should be described and at least some examples of R1, R2 and R3 for the current work should be given for clarity.
4. Similarly, I’d suggest addition explanations for the abbreviations in the tables.
My general feeling is positive, therefore I recommend publication with minor revision.
Author Response
Reviewer: 2
In the work entitled „Rhodium(III)-Catalyzed [4+2] Annulation via C-H Activation: Synthesis of Multi-substituted Naphthalenone Sulfoxonium Ylides „ by Xiaohan Song, Xu Han, Rui Zhang, Hong Liu, and Jiang Wangthe, the Authors touches an interesting method of [4+2] annulation of naphthalene derivatives on rhodium catalyst.
Generally, the work is presented clearly and contains a promising solution for obtaining substituted naphthalenes. There are also appropriate literature citations and the results are supported by relevant data.
Question 1: English revision recommended.
Response: We are very grateful to the referee for his/her suggestions.
We have revised English grammar and spelling in the revised manuscript.
Question 2: I wonder if the Authors did identify any other regioisomers in the reaction mixture among the products shown on Scheme 2. If yes, such information would be valuable in the method description.
Response: We are very grateful to the referee for his/her suggestions.
We didn’t identify any other regioisomers in the reaction mixture among the products shown on Scheme 2.
Question 3: Scheme 1: Abbreviations like DG, ACN should be described and at least some examples of R1, R2 and R3 for the current work should be given for clarity.
Response: We are very grateful to the referee for his/her suggestions.
We have described the abbreviations and given the clarity of R1, R2 and R3 in the revised manuscript.
Question 4: Similarly, I’d suggest addition explanations for the abbreviations in the tables.
Response: We are very grateful to the referee for his/her suggestions.
We have described the explanations for the abbreviations in the revised manuscript.
Reviewer 3 Report
In the manuscript titled, “Rhodium (III)-Catalyzed [4+2] Annulation via C-H Activation: Synthesis of Multi-substituted Naphthalenone Sulfoxonium Ylides”, the authors developed an efficient method for the synthesis of naphthalenone sulfoxonium ylides. The authors developed an efficient protocol under very mild reactions. Samilar work was published in Organic Letters on April 8th 2019 by Fan and co-workers (Org. Lett. 2019, 21, 2541−2545). However, in the current work, the conditions were mild (reactions were conducted at room temperature) and the products were obtained in high yields. Whereas in the work of Fan and co-workers, the yields of the products were very low and the reactions were conducted at high temperature. Fan and co-workers did not use any additives, where as in the current work, the authors used additives which is crucial for obtaining the products in high yields. In addition, the authors conducted the kinetic studies to determine the rate determining step. This work is good to publish in the journal Molecules. I had few concerns and comments that were listed below. I request the authors to address them before the manuscript getting accepted for publication.
1. In the proposed mechanism, in the first step, I am wondering that, is the conjugate base SbF6 is basic enough to abstract hydrogen? If so, then what is the role of Zinc acetate? Why both were required in the reaction? It would be great if the authors could explain this in the manuscript which facilitates the better understanding of the reaction to the reader.
2. The authors mentioned about the work of Fan and co-workers (ref. 52). But in the reference, the page numbers, year and volume were not mentioned. Please include the correct citation as the manuscript is already online.
3. There few typos in the manuscript. Please go through the entire manuscript and address these wherever necessary.
Author Response
Reviewer: 3
Comments:
In the manuscript titled, “Rhodium (III)-Catalyzed [4+2] Annulation via C-H Activation: Synthesis of Multi-substituted Naphthalenone Sulfoxonium Ylides”, the authors developed an efficient method for the synthesis of naphthalenone sulfoxonium ylides. The authors developed an efficient protocol under very mild reactions. Samilar work was published in Organic Letters on April 8th 2019 by Fan and co-workers (Org. Lett. 2019, 21, 2541−2545). However, in the current work, the conditions were mild (reactions were conducted at room temperature) and the products were obtained in high yields. Whereas in the work of Fan and co-workers, the yields of the products were very low and the reactions were conducted at high temperature. Fan and co-workers did not use any additives, where as in the current work, the authors used additives which is crucial for obtaining the products in high yields. In addition, the authors conducted the kinetic studies to determine the rate determining step. This work is good to publish in the journal Molecules. I had few concerns and comments that were listed below. I request the authors to address them before the manuscript getting accepted for publication.
Question 1: In the proposed mechanism, in the first step, I am wondering that, is the conjugate base SbF6 is basic enough to abstract hydrogen? If so, then what is the role of Zinc acetate? Why both were required in the reaction? It would be great if the authors could explain this in the manuscript which facilitates the better understanding of the reaction to the reader.
Response: We are very grateful to the reviewer’s comments.
(1) According to work of Cheng and co-workers (Org. Lett. 2018, 20, 5, 1396-1399), Cp*Rh(OAc)2 is more powerful than [Cp*Rh(MeCN)3)](SbF6)2 as a catalyst in the cross-coupling reaction of sulfoxonium ylides. And we speculate that Cp*Rh(OAc)2 is formed in our reaction after adding the Zinc acetate.
(2) In this reaction, AgSbF6 is a appropriate Ag salt to react with Cp*RhCl2 to enable the Rh(III) to act as a catalyst. And the role of Zinc acetate maybe abstract hydrogen.
(3) We have add a brief explanation for the function of Zinc acetate in the revised manuscript.
Question 2: The authors mentioned about the work of Fan and co-workers (ref. 52). But in the reference, the page numbers, year and volume were not mentioned. Please include the correct citation as the manuscript is already online.
Response: We are very grateful to the reviewer’s comments.
We have revised the form of reference 52 in the revised manuscript.
Question 3: There few typos in the manuscript. Please go through the entire manuscript and address these wherever necessary.
Response: We are very grateful to the reviewer’s comments
We have revised English grammar and spelling in our revised manuscript.